

# Behavioral responses of a clonal fish to perceived predation risk

Jonathan Aguiñaga, Sophia Jin, Ishita Pesati and Kate L. Laskowski

Ecology and Evolution, Center for Population Biology, University of California, Davis, Davis, United States of America

## ABSTRACT

Predation threat is a major driver of behavior in many prey species. Animals can recognize their relative risk of predation based on cues in the environment, including visual and/or chemical cues released by a predator or from its prey. When threat of predation is high, prey often respond by altering their behavior to reduce their probability of detection and/or capture. Here, we test how a clonal fish, the Amazon molly (*Poecilia formosa*), behaviorally responds to predation cues. We measured aggressive and social behaviors both under 'risk', where chemical cues from predatory fish and injured conspecifics were present, and control contexts (no risk cues present). We predicted that mollies would exhibit reduced aggression towards a simulated intruder and increased sociability under risk contexts as aggression might increase their visibility to a predator and shoaling should decrease their chance of capture through the dilution effect. As predicted, we found that Amazon mollies spent more time with a conspecific when risk cues were present, however they did not reduce their aggression. This highlights the general result of the 'safety in numbers' behavioral response that many small shoaling species exhibit, including these clonal fish, which suggests that mollies may view this response as a more effective anti-predator response compared to limiting their detectability by reducing aggressive conspecific interactions.

# INTRODUCTION

Predation threat is a major driver of behavior in many prey species. Grouping behaviors are often used as a way to significantly reduce per-capita predation risk (*Balaban-Feld et al., 2019a*; *Creel, Schuette & Christianson, 2014*; *Patin et al., 2019*; *Sih, 1997*; *van Langevelde, Suselbeek & Brown, 2022*; *Walls, 1995*). Grouping with others reduces predation risk by increasing overall vigilance (the 'many eyes' hypothesis; *Lima & Dill, 1990*) and also by reducing the odds of a given individual's capture (the dilution and confusion effects; *Lima, 1995*; *van Langevelde, Suselbeek & Brown, 2022*). Shoaling behavior in many fish species is a classic example of this anti-predator response. However, the density and cohesion of such groups are highly fluid as individuals often modulate their social tendencies in response to differences in internal states such as nutritional condition (*Balaban-Feld et al., 2019b*; *Conradt & Roper, 2000*) and environmental cues (*Jolles et al., 2017*; *Morrell et al., 2008*; *Webster et al., 2013*).

Corresponding author
Jonathan Aguiñaga,
jaguinaga@ucdavis.edu

Close proximity to others, as is typical in shoaling fish, can also have negative impacts on individuals (*Kelley et al., 2011*). For example, group-living individuals can often experience high exploitation competition which can lead to differential foraging success and variable levels of aggression within a group (*Balaban-Feld et al., 2019a*; *Melotto, Ficetola & Manenti, 2019*). Aggression is a costly behavior to engage in not just because of risk of injury (*Oliveira, Silva & Simões, 2011*; *Teles & Oliveira, 2016*) but also because it can simultaneously decrease vigilance and make the individuals more conspicuous to potential predators (*Kim, Brown & Grant, 2004*; *Pintor, Sih & Bauer, 2008*). For example, juvenile salmon became less aggressive towards a mirror image when bird predator chemical cues were added to the water (*Martel & Dill, 1993*). Pairs of fighting cichlids switched from high-intensity aggressive behaviors (mouth-wrestling) to low-intensity behaviors (lateral displays) when a model predator was introduced (*Brick, 1998*). Altogether there is considerable evidence that individuals modulate their social and aggressive behaviors while under predation risk and that this can often depend on the intensity and frequency of perceived risk (*i.e.,* risk-allocation and threat-sensitive hypotheses; *Brick, 1998*; *Brown et al., 2006*; *Ferrari, Messier & Chivers, 2008*; *Ferrari et al., 2010*). Social prey must therefore balance the energetic demands for acquiring resources and defending them from others with the potential for such behaviors to increase their visibility to predators. As such, there should be strong pressure for individuals to modulate such behaviors relative to their perceived risk of predation.

A general finding in many fish species is that individuals should increase their tendency to associate with conspecifics and reduce their aggression towards an intruder in the presence of predator cues (*Herbert-Read et al., 2017*; *Kelley et al., 2011*; *Martel & Dill, 1993*). However, most of this work has been done on sexually reproducing species (reviewed in *Kelley & Magurran, 2003*) but there are reasons to suspect that these patterns may differ in clonal species, which are often under-used models for ecology and evolutionary research (*Laskowski et al., 2019*). In clonally reproducing species, individuals are highly related and kin selection theory predicts that relatedness can be used to explain changes in behavior, particularly the existence of greater cooperativeness and even seemingly altruistic behaviors (*West, Griffin & Gardner, 2007*). In regard to anti-predator behaviors, when individuals group with other clones of themselves, the costs of competition and predation may be dampened if kin selection trumps the effects of individual selection (*Griffin & West, 2002*; *West, Pen & Griffin, 2002*). As such, we may expect clonal individuals to exhibit dampened anti-predator behavior. On the other hand, clonal individuals are still subject to heavy predation and must still compete with each other for resources and as such are subject to the same trade-offs between competition and anti-predator behavior as any other sexually reproducing species.

Here we aim to understand the anti-predator behavioral responses in a naturally occurring clonal fish, the Amazon molly (*P. formosa*). The Amazon molly is small, live-bearing, freshwater fish in the Poecilid family. The unisexual Amazon molly originated from a single hybridization event between a male sailfin molly (*Poecilia latipina*) and female Atlantic molly *(Poecilia mexicana)* approximately 100,000 years ago (*Lampert & Schartl, 2008*; *Schartl et al., 1995*; *Schlupp, Parzefall & Schartl, 2002*; *Stöck et al., 2010*;

*Tiedemann et al., 2005*). The Amazon molly reproduces gynogenetically, which requires her to use the sperm of closely related Poeciliids to initiate embryogenesis, though the genetic material within the sperm is not incorporated into the ova (*Makowicz et al., 2022*). Thus, Amazon mollies produce broods of offspring that are identical to each other and the mother (*Schartl et al., 1995*). There is evidence that Amazon mollies can use both visual and chemical cues to discriminate between sister, non-sister clones, and heterospecific individuals, which is a major pre-cursor assumption for kin selection to operate (*Makowicz et al., 2016*; *Makowicz, Moore & Schlupp, 2018*). Given these findings, it is possible that Amazon mollies may differ in their anti-predator responses from previously established work in sexually reproducing species (*Balaban-Feld et al., 2019a*; *Blake et al., 2015*; *Herbert-Read et al., 2017*; *Kelley et al., 2011*; *Magurran, 1990*; *Magurran & Seghers, 1991*). Specifically, we tested whether and how Amazon mollies altered their expression of aggressive and social behaviors under short-term increases of perceived predation risk compared to control conditions where no additional cues of risk were present. All fish exude chemical cues into the surrounding water and there is strong evidence that prey species recognize and respond to the chemical cues of predatory fish (*Blake et al., 2015*; *Brown, Paige & Godin, 2000*; *Ferrari, Capitania-Kwok & Chivers, 2006*; *Holmes & McCormick, 2010*). Additionally, many prey species contain so-called 'alarm cues' in their skin which are released when the skin is damaged (*Wisenden, 2011*; *Wisenden, 2014*; *Wisenden et al., 2009*). There is also strong evidence that many small fish species use these chemical cues, both from the predators and injured conspecifics, to help gauge their current risk of predation (*Herbert-Read et al., 2017*; *Magurran, 1990*; *Magurran & Seghers, 1991*). As such, these were the cues we used to manipulate perceived predation risk in our mollies.

## MATERIALS & METHODS

### Study system

For our experiments, which we detail below, we use a naturally occurring lineage of Amazon mollies that were established from a single female collected from the wild (Weslaco, Texas) in 2015, approximately 10–20 generations ago (collected by Amber Makowicz, labeled as '3N' lineage, personal communication).

### Animal husbandry

All lab stocks of fish are maintained in a vivarium located in our laboratory at the University of California Davis. The system comprises nine independent racks; within a rack, water re-circulates after passing through mechanical, biological and carbon filters. 10% of the water is exchanged by a daily automated exchange. Stocks of fish are maintained in large (75 liter or 113 liter) tanks in mixed-age groups throughout their lives (at densities of 1–2 fish/3.8 liters), on a 14:10 L:D light cycle, and maintained at 28C through sump heaters. Water for our all our aquaria is produced through a reverse osmosis system and water quality (pH, conductivity, temperature) is automatically monitored and adjusted as necessary to maintain within appropriate bounds. Fish health and welfare are checked
daily, and fish are fed *ab libitum* twice daily with a mix of tropical fish flakes (Tetramin), thawed frozen bloodworms and newly hatched Artemia.

We selected 30 adult individuals as our experimental animals from these stock tanks and placed them into three 37.8-liter tanks (10 fish per tank; also maintained on the same water monitoring system) where they remained for the duration of the experiment. At the time of removal from the stock tanks, these fish were subcutaneously marked with Visible Implant Elastomer Tags (VIE Tags; Northwest Marine Technology, Inc., Anacortes, WA, USA) under anesthesia with MS-222 (150 mg/l) in pH-buffered (with sodium bicarbonate 1.3 g/l) RO water. Individuals recovered for a week before the start of experiments.

During the experiment, all fish swam normally, showed no signs of erratic or abnormal behaviors or evidence of disease. At the conclusion of these experiments, all fish were returned to their group housing tanks.

## Chemical cue preparation

To generate the predator cues, we used the electric yellow lab cichlid (*Labidochromis caeruleus*). Each predator was housed independently from the molly rack system in individual 151.4-liter tanks using the same RO water source and water quality monitoring system as our main molly system. Water was filtered through a mechanical and biological filter and 10% of the water is manually exchanged weekly. Cichlids were fed twice daily *ab libitum* with cichlid pellets (Omega One). While the yellow lab cichlid is not sympatric with Amazon mollies, these cichlids are still effective predators on many small fish species and will readily consume juvenile mollies under lab conditions (J Aguiñaga, 2021, pers. obs). There is considerable evidence that many small shoaling fishes, including Poecilids, exhibit generalized responses to many novel predators (*Blake et al., 2015*; *Brown et al., 2013*; *Rehage, Barnett & Sih, 2005*; *Swaney, Cabrera-Álvarez & Reader, 2015*). To ensure that the mollies recognize the predator cues as dangerous, we paired these cues with alarm cues from injured conspecifics. Several studies have demonstrated that pairing chemical predator cues with injured conspecific cues is successful at eliciting anti-predator responses and encouraging learning of predator chemical cues as indicative of increased predation risk (*Brown et al., 2013*; *Holmes & McCormick, 2010*; *Korpi & Wisenden, 2001*; *Larson & McCormick, 2005*; *Swaney, Cabrera-Álvarez & Reader, 2015*). We extracted predator odor cues by removing water from one of two 151.4-liter tanks which housed one yellow lab cichlids each (protocol adapted from *Ylönen et al., 2007*). At the start of each testing day, 200 ml was taken from one of these holding tanks (alternating between days) and used for the daily trials.

To simulate high risk, we extracted chemical alarm cues from conspecific Amazon mollies and paired them with predator odors from yellow lab cichlids. We prepared chemical alarm cues from freshly deceased (<15 h) Amazon mollies that showed no signs of decomposition (*Wisenden et al., 2009*). These fish were collected immediately when they were spotted in lab stock tanks through daily routine health checks of our lab populations. To prepare these alarm cues, we scored freshly deceased fish on their flanks (5 scores each side) with a razor blade (as in *Chivers et al., 2014*; *Wisenden, 2011*). Each damaged fish was dipped into 50ml of deionized water and swirled around for 5 min. Then, 10ml of this

solution was pipetted into ice cube aliquots and frozen for future trials (protocol adapted from *Crane & Ferrari, 2015*), providing an estimated concentration of 2 cuts per 10 ml. Previous work indicates that chemical alarm cues do not show signs of degradation until they are mechanically removed from the skin and that freezing freshly extracted chemical cues retains their potency (*Wisenden et al., 2009*). For each of the trials involving predation risk, we simulated high risk by pouring in the chemical alarm cues and predator odors into the arena. Hereafter, we will refer to the pairing of chemical alarm cues and predator odors as 'risk cues'.

## Aggression trials

We measured the aggressive behavior of individual Amazon mollies under two contexts: with risk cues ($n = 10$) or without risk cues (*i.e.,* control, $n = 10$), totaling to 20 unique fish tested. Fish were randomly assigned a treatment group and remained in that treatment group for the entire experiment. Each fish was tested between 1000–1600 h from June–July 2022 for two repeated trials which were separated by 8 days. Over the course of the experiment, one fish per treatment died (total $n = 18$, or $n = 9$ per treatment). Experimental animals were all at least 1 year old adults and mollies typically survive for 1–2 years in laboratory conditions; there was no indication of any disease or abnormal behavior in any of our remaining animals.

The experimental arena consisted of one half of a 37.8-liter tank (49 cm × 25 cm × 28 cm; Fig. 1A). The long side of the tank was lined with mirror tape and adhered with aquarium safe glue. We lined all internal walls with coroplast cutouts to prevent the fish from observing the outside environment and to remove reflective surfaces other than the mirror tape placed on the long inner wall. Tanks were lined with white gravel and filled with water (24C) to a height of 10 cm. The shelves on which the experimental tanks were placed were blinded with curtains to limit outside disturbance. Tanks were illuminated by overhead fluorescent lights which were diffused through cloth to limit glare. Air stones actively pumped fresh air into the tank but were turned off 5 min prior to the trial.

We followed the protocol from *Oliveira, Silva & Simões (2011)* and measured aggression using a mirror assay. Each fish was individually placed into the testing arena and allowed to acclimate for 24 h prior to the trial; fish were not fed during this time. After this acclimation period and at the start of the observation, we added 10 ml of alarm cues and 25 ml of predator odor cues for individuals in the predation risk treatment. Fish in the control treatment were given blank water cues (10 ml then 25 ml). We removed the mirror cover three minutes after the risk or blank water cues were introduced into the arena. Trials lasted 5 min and we recorded each fish using an overhead webcam (Logitech C920e) connected to a laptop computer. After the trial concluded, the focal fish was returned to a group housing tank and the experimental arena was thoroughly washed with tap water to rinse off any residue chemical alarm cues. The arena was allowed to dry for 24 h before being used again. Each animal was tested again 8 days later in the same context. We recorded the number of bites that each fish directed towards its reflection as our measure of aggression (*e.g.,* *Way et al., 2015*). In these fish, bites resemble pecking motions. We can observe bites by monitoring the rapid motions of the jaws and head of the focal animal. This usually

a.

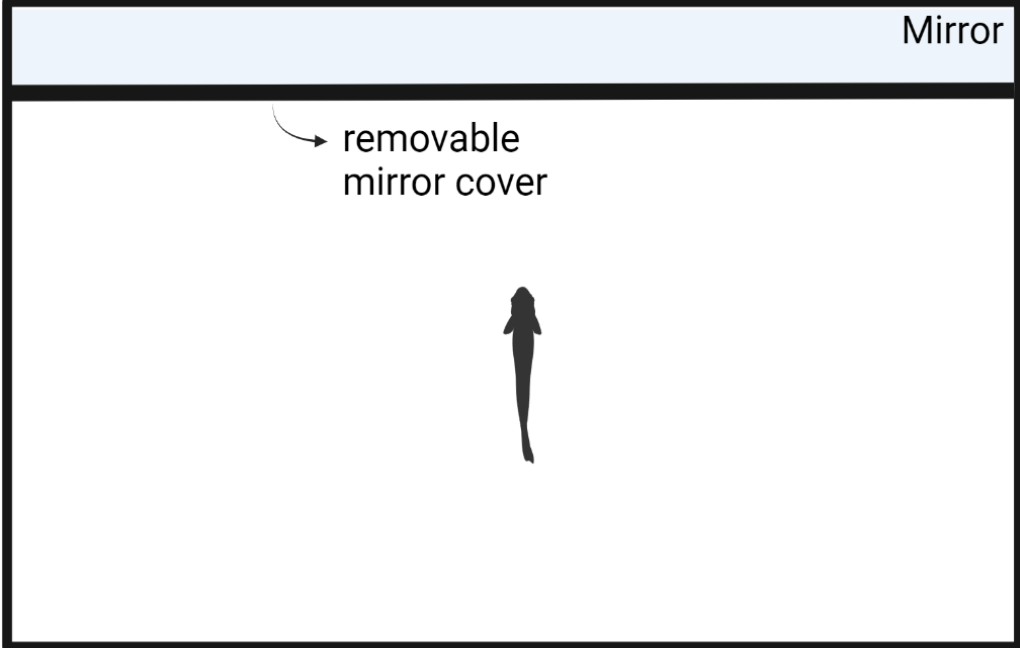

b.

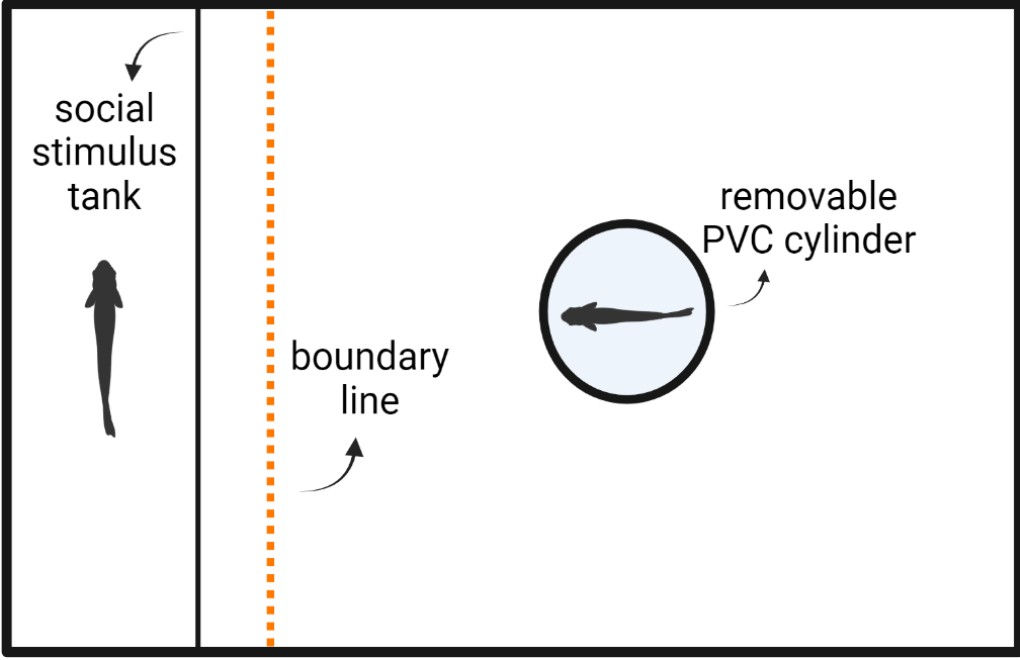

**Figure 1  Experimental arenas for testing (A) aggression and (B) sociability behaviors.** After blank water cues or risk cues were introduced into the environment, a cover which obscured the mirror in (A) and the opaque PVC cylinder in (B) were removed, after which the trial begins. Created with https://www.biorender.com/.

indicates an aggressive encounter as fish receiving the aggression may dart away, and this has been used as measure of aggression in previous work on this species (*Laskowski, Wolf & Bierbach, 2016*; *Makowicz, Moore & Schlupp, 2018*). Bites were manually scored using BORIS (*Friard & Gamba, 2016*).

## Sociability trials

We measured the social tendencies of Amazon mollies ($n = 10$) under two contexts: with risk cues or without risk cues (*i.e.,* control). These were a separate cohort of fish than those that were measured for aggression as we were concerned that re-using the same fish from the aggression trials could result in carry-over effects from the previous exposures to the risk cues. Each individual was measured three times in each context, once every other day. We tested each fish in the control trials first (*i.e.,* trials 1–3) and then in the risk cue trials (4–6) as we expected the risk cues to have the potential for a larger carryover effect on later behavior than the control trials would have on later behavior. After each trial, the focal fish were returned to a group housing tank and the tanks were cleaned to remove chemical cues from the predator and conspecific as was done in the aggression trials.

We used a 37.8-liter fish aquarium ($49 \times 25 \times 28$ cm) as our experimental arena. We placed a transparent plastic fish carrier tank ($22 \times 14 \times 12$ cm) next to one of the inner walls inside of the experimental arena (Fig. 1B). Both the carrier tank and experimental arena were lined with white gravel and filled with water (24C) to a height of 10 cm. Experimental arenas were blinded, illuminated, and aerated as in the aggression assay.

We measured sociability as the tendency for an individual to associate with a conspecific, similar to measurements used in other studies (*Gartland et al., 2022*). Prior to the start of a trial, we placed a size-matched conspecific fish (±three mm) into the plastic carrier tank within the experimental arena to act as the social stimulus fish. Water between the carrier tank and the experimental arena was not shared preventing the transfer of any chemical cues. The focal fish was then placed into an opaque PVC cylinder within the arena to acclimate for five minutes. During these five minutes, blank water cues or risk cues were added into the experimental arena as in the aggression assay. At the end of these five minutes, the PVC cylinder was manually removed, and the 15-minute trial began. We recorded the trials using an overhead webcam (Logitech C920e) connected to a laptop computer. We recorded the total amount of time in seconds out of the entire duration of the trial (*i.e.,* 900s) that the focal individual spent within six cm of the stimulus fish (six cm equates to approximately two body lengths) using BORIS (*Friard & Gamba, 2016*).

## Statistical analysis

We used general linear mixed models to test how molly aggressive and social behavior responded to risk cues. We ran a separate model for each of our behavioral variables (total number of bites for aggression; total time spent near conspecific for sociability). Each model included the fixed effects of risk context (control or risk cues present), trial (coded as a continuous variable within each context, *i.e.,* 1–2 for the aggression assays and 1–3 for the sociability assays) and the interaction between these two terms. We also included the random effect of individual to account for repeated measures. In both models, we first

tested the significance of the interaction term (trial x context), which was found to be non-significant in both cases (see Results), therefore to increase our statistical power to detect effects in our small sample size (*Zuur et al., 2009*), we re-fit the models excluding the interaction to test for the main effects of context and trial. We used $F$-tests with Satterthwaite's method of degree of freedom estimation for significance testing of the fixed effects and log likelihood ratio tests for the random effect. All models were run with the statistical computing language R (*R Core Team, 2020*) using the 'lme4' package (*Bates et al., 2014*); $F$-tests were performed using the package 'lmerTest' (*Kuznetsova, PB & Christensen, 2017*) and R-squared values were estimated with 'MuMIn' (*Barton & Barton, 2015*). For each model, we visually inspected the residuals to ensure we met model assumptions of residual homogeneity of variance and normality, which were met in both cases.

### Ethics statement

All husbandry and experimental protocols are approved by UC Davis's Institutional Animal Care and Use Committee (IACUC; Protocol #21897).

## RESULTS

The mirror test was successful at eliciting aggression from the Amazon mollies (Fig. 2A). However, we found no evidence that Amazon mollies modulated their aggression in response to the presence of risk cues. There was no significant interaction between context and trial ($F_{1,32} = 0.10$, $p = 0.75$) nor any overall differences between contexts ($F_{1,33} < 0.001$, $p = 0.99$) or over trials ($F_{1,33} = 0.17$, $p = 0.68$). We also found no evidence that individuals exhibited consistent individual differences in their aggressive behavior across the two trials (LRT = 0, $p = 1$).

   Amazon mollies exhibited considerable sociability spending the majority of the trial time (15 min) within two body lengths of the conspecific (Fig. 2B). There was no interaction between predator context and trial ($F_{1,47} = 0.37$, $p = 0.54$); however mollies spent significantly more time with the conspecific in the presence of risk cues (effect of risk cue = 103.44 s, $F_{1,48} = 8.22$, $p = 0.006$) and less time overall with the conspecific across the repeated trials (effect of trial = −51.44 s, $F_{1,48} = 5.42$, $p = 0.02$). We did see evidence of consistent individual differences in social behavior ($R = 0.25$, LRT = 5.95, $p = 0.01$). Results for these experiments are summarized in Table 1.

## DISCUSSION

Predators are an important influence on prey animals' behavior. Here, we show that the clonal Amazon molly responds to heightened perceived predation risk by increasing their sociability and spending more time near a conspecific. However, counter to our expectations, these mollies did not alter their aggressive behavior towards a simulated intruder, rather, they were relatively aggressive regardless of the risk context. Our work demonstrates that despite their clonal nature, Amazon mollies respond to predation in similar ways as most sexually-reproducing small, shoaling fishes, and likely view shoaling behavior as a more effective anti-predator response.

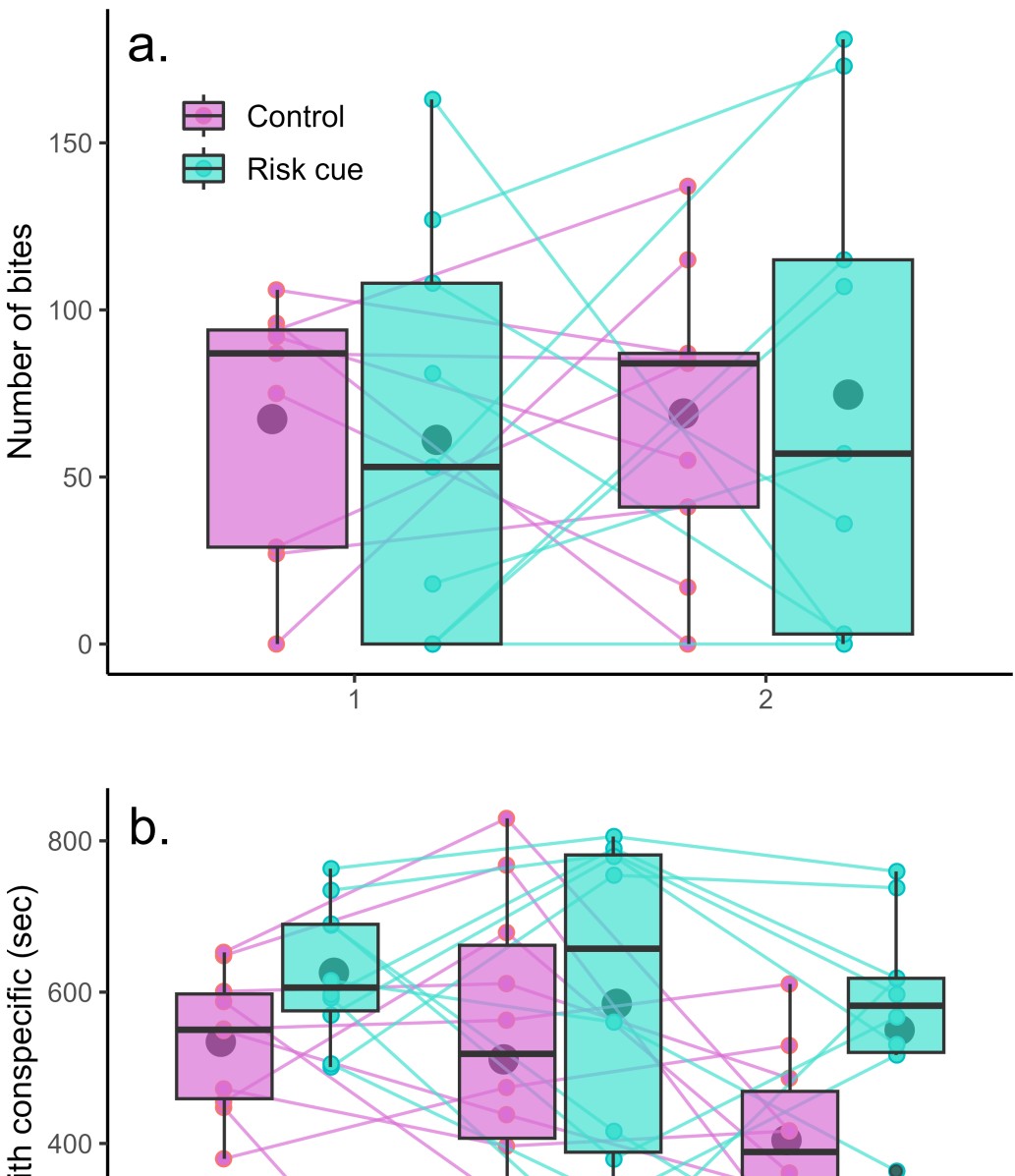

**Figure 2   Behavioral responses to risk and control conditions.** Amazon mollies do not modulate their (A) aggressive behavior but do increase time spent with a (B) conspecific in response to predation threat. Each point represents a single trial; lines connect repeated measurements on the same individuals. Box-plots show the responses within a given trial and context where the center line is the average and edges of the box delineate the inter-quartile range.

**Table 1  Model outputs for aggression and sociability experiments.** Results of general linear mixed models testing the effects of risk context and trial on behavioral metrics. Fish ID was included as a random effect. Test-statistic is either an F-statistic for fixed effects, or a log likelihood ratio test for variance components.

| Effect | Estimate (std. error) | Test-statistic (df) | *p*-value |
|---|---|---|---|
| Aggression: number of bites ($R^2_{marginal} = 0.005$; $R^2_{conditional} = 0.005$) | | | |
| *Intercept* | 56.75 | | |
| Trial | −7.61 (18.36) | 0.17 (1, 33) | 0.68 |
| Context (Risk cues) | 0.28 (18.36) | <0.001 (1, 33) | 0.99 |
| ID variance | 0 | 0 (1) | 0.99 |
| Residual variance | 3037 | | |
| Sociability: time spent near conspecific ($R^2_{marginal} = 0.15$; $R^2_{conditional} = 0.36$) | | | |
| *Intercept* | 586.15 | | |
| Trial | −51.44 (22.09) | 5.42 (1, 48) | 0.02 |
| Context (Risk cues) | 103.44 (36.07) | 8.22 (1, 48) | 0.006 |
| ID variance | 6676 | 5.96 (1) | 0.01 |
| Residual variance | 19519 | | |

The unique reproductive strategy of Amazon mollies means that each female can produce broods of offspring that are genetically identical to each other and their mother (*Lampert & Schartl, 2008*; *Schartl et al., 1995*; *Schlupp, Parzefall & Schartl, 2002*). On the one hand, this clonality presents the opportunity for strong kin selection which could offer an explanation for why Amazon mollies might show reduced anti-predator responses. A molly surrounded by clones may respond to risk differently than one in a more genetically heterogenous environment. Previous work has shown that Amazon mollies can recognize their sister clones and become more aggressive towards different clonal lineages and heterospecifics (*Makowicz et al., 2016*; *Makowicz, Moore & Schlupp, 2018*). Wild populations of mollies consist of multiple lineages presenting the opportunity for mollies to show preferences for associating with particular lineages, though this has never been formally tested. This could present exciting opportunities for future work: do mollies show increased anti-predator responses in genetically heterogenous social groups compared to when surrounded by sister clones? On the other hand, the Amazon molly, as is the case with all unisexual vertebrates, is a 'frozen F1 hybrid' (*Laskowski et al., 2019*) of its two parental species, the Atlantic and sailfin mollies (*Lampert & Schartl, 2008*; *Schartl et al., 1995*; *Tiedemann et al., 2005*). As such, the Amazon mollies genome is the combination of two sexually reproducing species and thus has not evolved clonality through natural selection (*i.e.,* slowly over evolutionary time). This means, the evolutionary forces that shaped the Atlantic and sailfin mollies' genomes are still apparent in the Amazon mollies' genomes and so we should expect similar anti-predator responses in this clonal fish as in other sexually reproducing small shoaling fishes. Our results indicate that Amazon mollies do recognize and respond to chemical cues indicating increased risk (conspecific alarm cues and novel predator cues) by increasing their preference to be near conspecifics, which is the expected response of shoaling fishes.

In many small fish species, shoaling and schooling behaviors have arisen as a response to reduce predation risk (*Magurran, 1990*; *Morgan & Godin, 1985*; *Seghers, 1974*). Large

groups can thus exhibit greater collective vigilance and overall reduced probability of individual prey capture through the dilution and confusion effects (*Creel, Schuette & Christianson, 2014*; *Lima, 1995*). As expected, in our study, the Amazon mollies spent more time near a conspecific when alarm and novel predator odor cues were added to their environment, increasing their perception of predation risk. Interestingly, there was also a general decline in sociability over time (trial number) which could be indicative of an overall habituation effect. And while the interaction between trial number and predator context was not significant, it does appear that there may be trend for a stronger decline in this sociability in fish tested under the control treatment (Fig. 2B). To explore this, we decided to perform a *post-hoc* analysis by testing for the effect of trial within each context separately. Indeed, in control conditions there is a suggestive trend for a decline in sociability over time (effect of trial $= -65.03 \pm 31.45$, $F_{1,19} = 4.28$, $p = 0.052$) whereas fish measured under the risk context exhibit no such habituation (effect of trial $= -37.8 \pm 32.6$, $F_{1,19} = 1.34$, $p = 0.26$). While this should be interpreted very cautiously as it is a post-hoc analysis, such a decline under control conditions could be evidence of habituation and might be expected as there is no apparent benefit to grouping with a conspecific in this context. The fact that when presented with risk cues, fish maintain their levels of sociability over the repeated trials further supports that our experimental fish were interpreting the alarm and predator cues as actual increases in their predation risk in this sociability assay.

In contrast to the predicted increase in sociability in the response to risk, the Amazon mollies do not appear to modulate their aggression even in the face of increased perceived predation risk. Instead, the mollies are consistently relatively aggressive regardless of their treatment. Amazon mollies are known to fight with each other for dominance and these fights can continue until an individual exhibits submissive behaviors (*Laskowski, Wolf & Bierbach, 2016*; *Makowicz et al., 2016*; *Makowicz, Moore & Schlupp, 2018*). Here, the mirror reflection prevents this which may have heightened the focal individual's motivation to continue interacting with their reflection even in the face of (uncertain) danger. This is further supported by the fact, that anecdotally, we did not see much evidence of aggressive behavior directed towards the live conspecific in the sociality assay.

Another explanation for the lack of any changes in aggressive behavior may be that the presence of chemical cues and lack of visual predator cues could be perceived as relatively low risk and uncertain danger. While chemical cues are indicative that predators may be in the area, they present more uncertain information about the immediate danger than, for example, a visual cue might (*Crane & Ferrari, 2016*; *Stephenson, 2016*). For many aquatic organisms, chemical cues can be used during foraging, mate choice, dispersal and migration, and species recognition behaviors (*Ward & Mehner, 2010*). When visual cues are present and align with chemical cues, the multimodal information likely provides more certainty about the state of the environment compared to when only unimodal cues are available (*Stamps & Bell, 2020*). For example, in sulphur mollies (*Poecilia sulphuraria*), when acoustic and visual cues of diving birds are present, mollies dive into deeper waters more quickly compared to when only one cue is present (*Lukas et al., 2021*). In other aquatic organisms, simultaneous presence of chemical and visual cues can modulate aggression, sociability, predator inspection and refuge use behaviors

(*Ocasio-Torres, Crowl & Sabat, 2021*; *Ward & Mehner, 2010*; *Wilson, TM & Ward, 2022*). It could be that the mollies in the aggression assay interpreted the risk cues in the absence of visual cues as a low-risk and thus valued placing more effort into aggressive interactions. If habitats have little or few predators, then it may be adaptive for individuals to ignore predation risk and attempt to usurp higher ranking individuals or competitors for important resources like territory, food patches, or mates (*Magurran & Seghers, 1991*). It is also possible that we did not see a change in aggressive behavior because the Amazon mollies were unable to detect the alarm and predator cues. This seems unlikely given that they did respond to these cues in the same concentrations in the sociability assay. This aligns with previous work which has shown that such cues are effective at eliciting behavioral responses in many small fish species (*Brown, Paige & Godin, 2000*; *Brown GE & Adrian Jr, 2004*; *Brown, Laland & Krause, 2011*; *Ferrari, Capitania-Kwok & Chivers, 2006*; *Ferrari & Chivers, 2006*; *Zhao, Ferrari & Chivers, 2006*).

## CONCLUSIONS

This study highlights the general patterns of anti-predator behaviors in small shoaling fish. Our work is consistent with previous work on other small sexually-reproducing shoaling fishes showing that under increased perceived risk, clonal Amazon mollies spend more time with conspecifics. This result supports the generality of how social prey animals increase time spent with others while under risk, a result that is widely documented in a variety of taxa. Further research is needed to evaluate how consistent long-term exposure to predation risk influences individual and groups of clonal Amazon mollies in similar social and aggression behaviors.

## ACKNOWLEDGEMENTS

We would also like to thank Karen Kacevas for assistance with fish husbandry.

### Funding

Funding was provided to Jonathan Aguiñaga from a student research grant from the UC Davis Center for Population Biology and by NSF IOS-2100625 to Kate L. Laskowski. The UC Davis Young Scholars Program, which provides high school students with the opportunities to conduct hands on research at the university level, provided funding to Jonathan Aguiñaga to buy supplies and materials. The funders had no role in study design, data collection and analysis, decision to publish, or preparation of the manuscript.

### Grant Disclosures

The following grant information was disclosed by the authors:
The UC Davis Center for Population Biology and by NSF IOS-2100625.
The UC Davis Young Scholars Program.

## Competing Interests

The authors declare there are no competing interests.

## Author Contributions

- Jonathan Aguiñaga conceived and designed the experiments, analyzed the data, prepared figures and/or tables, authored or reviewed drafts of the article, and approved the final draft.
- Sophia Jin conceived and designed the experiments, performed the experiments, analyzed the data, prepared figures and/or tables, and approved the final draft.
- Ishita Pesati conceived and designed the experiments, performed the experiments, analyzed the data, prepared figures and/or tables, and approved the final draft.
- Kate L. Laskowski conceived and designed the experiments, analyzed the data, prepared figures and/or tables, authored or reviewed drafts of the article, and approved the final draft.

## Animal Ethics

The following information was supplied relating to ethical approvals (i.e., approving body and any reference numbers):

UC Davis' Instututional Animal Care and Use Committee (IACUC) provided full approval for this research.

## Data Availability

The raw data measurements and code used to process these measurements are available in the Supplementary Files.

## Supplemental Information

Supplemental information for this article can be found online at http://dx.doi.org/10.7717/peerj.17547#supplemental-information.

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
