# Peer review of "Behavioral responses of a clonal fish to perceived predation risk"

_PeerJ, doi:10.7717/peerj.17547_

## Round 0.1 · original submission · Major Revisions

Dear Jonathan,

Thank you for submitting this interesting study to PeerJ. I have now received two referee reports that are generally quite positive, but both offer constructive criticisms that should be taken into consideration prior to acceptance of your manuscript. I particularly agree with the comment from Reviewer 2 that it would be helpful to expand on the theory and rationale in the introduction for why clonal fishes may have different predator responses compared to sexually reproducing species.

I look forward to reading a revised version of your manuscript. Please note that PeerJ does not provide copyediting as a standard service, so I encourage you to carefully proofread the paper prior to submitting a revised version. For example, I noticed several inconsistencies with your reference formats that should be corrected (e.g. sometimes including the first names or initial(s) of authors, inconsistent use of commas after "et al").

Sincerely,
Andy

Reviewer 1 ·

Basic reporting

In this study, the authors investigated two elements of behavioral responses to perceived predation risk in a clonal molly: aggression and sociability. They found that aggression performed towards a mirror was no different, whereas fish spent more time near a live conspecific when predation risk was elevated. While the experiments were relatively simple, the results are clear and are consistent with many other studies of predation responses in fishes. Hopefully, these results will serve as a baseline for interesting variants of this question in the future. Overall, I have only a few comments that I believe should be addressed to improve the strength of the manuscript. Please see below.

Experimental design

See comments below.

Validity of the findings

See comments below.

Additional comments

Introduction

Line 12: This opening line seems a little generic. Predation is a major threat to all living things, why make it specific only to shoaling fish?
Line 23: ‘universality’ seems a little too strong here. While this is generally true, sometimes grouping preferences of fish decrease under predation threat (e.g., 10.1016/j.beproc.2021.104464).
Line 31: Not certain what the citation style for this journal is, but seems unnecessary to list 6 names before et al.?
Line 36: The classification of a species as ‘shoaling’ is based on whether it is usually found in a group. Therefore, it seems redundant to state this since this is the definition of shoaling?
Line 38: Worth mentioning that internal state can also affect social tendencies, not just changes in external environment?
Line 61: It would be better to describe the reasons why you suspect this immediately after this sentence, and then get into specifics about your model species afterwards. It currently reads like a cliff-hanger and disrupts the flow of the paragraph a bit.
Line 70: How do they recognize them (vision, olfaction, etc.)? This info seems relevant since your sociality assay relied on vision and did not allow perception via olfaction.
Line 78: Define gynogenetically here instead of below.
Line 79: ‘…is a particularly useful system…’

Methods

Line 97: Approximately how many generations?
Line 102: How many fish were housed together in each tank? How old were they (juveniles or adults)?
Line 103: Conditioned with what? What was the source of the water (well water, tap water, etc)? How often was water quality checked and were water changes performed? Was temp maintained using a heater? Was filtration used? Supplemental aeration?
Line 104: What brand of fish flakes?
Line 107: Not clear whether the stated concentration is of MS-222 or the reagent used to buffer the water (which was what?). What pH was the water buffered to? Also, this concentration seems quite low; lethal doses are usually 200-500 mg/L while anesthetic doses are often 50-100 mg/L (e.g., 10.30802/AALAS-JAALAS-19-000067). I would double check that the written dose is correct.
Line 130: I would argue that 24h is not ‘freshly deceased’. Also, what evidence is there that alarm cues do not degrade/dissipate after a fish has been dead for several hours? Your latter comments about cues being less effective when frozen 3-6h after collection vs. frozen fresh raise questions about this. Better justification of the methods used is needed in this aspect.
Line 157: 10% mortality across an 8-day period seems rather high, no? Is there a reasonable cause attributed to this (infection from injections etc)?
Line 177: Washed with what?
Line 187: Missing period
Line 201: Provide an estimate of how closely size-matched fish were. Also, was the side of the tank that the conspecific was placed on counter-balanced across trials and/or individuals?
Lines 208-211: The issue with these types of grouping assays is that fish can be close in proximity because of prosocial reasons (grouping) or anti-social reasons (aggression). I see that you anecdotally state that rates of aggression were low between conspecifics (side note; this is unusual given the high rates of aggression seen during the mirror assay. Why might this difference have occurred?), but it would be much more convincing to have actual aggression data to support the prosocial nature of time spent with a conspecific during this assay.
Line 217: Can you briefly state how the ‘trial’ effect was structured in terms of levels? One would assume something like ‘1st, 2nd, 3rd’ within each treatment (instead of 1-6), but not certain as currently written. (I know your code/data are provided, but easier for readers if clearly stated here.)

Results

Line 242: Consider rewording “more than a majority” to “the majority”. A majority includes anything from 51-100%, so not clear how something can be more than this?

Discussion

Lines 288-295: This section of the discussion isn’t supported by the stats that you currently report (e.g., the interaction term was p=0.54). I would either remove this whole section or conduct supplemental post hoc analyses that better allow you to make these statements (e.g., separate LMMs across time for each treatment group individually). You still need to use caution when reporting the results of these post hoc analyses since they were not part of your initial analysis plan, but it would be more convincing than what is presented at present.
Lines 326-328: The fact that your lowest and highest average sociability scores were obtained during back-to-back trials (3 vs 4) where only the addition of the predation cue was different is also pretty good evidence for this!

Reviewer 2 ·

Basic reporting

- Introduction: The general introduction to this paper provides a good overall structure for background and context, however, some ambiguous passages could presented more clearly/more information would help to clarify the context, for example:
o Line 58: “the general prediction” – what is this a prediction referring to? References should also be included to support the statement made.
o The sentence beginning on line 60 and ending on line 61 needs reference(s).
o Statements from line 64 to 69 require references (and throughout the paper).
o The statement starting at the end of line 71 and ending on line 73 seems to suggest that the study will compare response to predator cues between sexually reproducing and non-sexually reproducing species. I suggest clarifying.
o The rationalization for the use of the clonal species on line 79 – 82 differs from the rationalization in the previous paragraph. I would suggest the authors revisit these rationalizations and provide a rationalization/context that most closely matches the experimental design.
o I would suggest the authors provide more context/background on perceived predation risk in the introduction as it relates to kairomones and alarm cue as this is the basis of the research
- Methods:
o The information provided on lines 88-90 seems to be a repeat of the information provided in the previous paragraph. I suggest removing this redundancy.
o It is unclear how the information provided relating to the 3N line (lines 90-96), while interesting, relate to the current study. I would suggest the authors either add a statement to clarify or remove this section.
o If possible, it would be beneficial to include the number (or estimate) of generations the study animals had been reared in captivity (lines 96-99).
o Animal Husbandry: I suggest the authors include more information about housing conditions – how many housing tanks were used? Number of individuals per tank? Where (location) were they housed? Lighting conditions? Recirculation system or flow-through, other? How long were they housed in the new facility prior to the start of experimentation?
o For better flow, I think paragraph from line 110-122 should be included in the following section ‘Chemical cue preparation’
o More information about the housing of the predator should be included to assess how predator cue was collected (see for example Brown and Smith, 1998 Can. J. Fish. Aquat. Sci. 55: 611-617 )
o It is unclear what the statement on line 120 “…as indicative of increased risk” is referring to. I would suggest clarifying.
o Statements such as "All fish in this study were healthy…” (Line 124) is a little ambiguous – I would suggest avoiding making such a claim unless it is made in reference to a specific target or definition of “healthy”
o I suggest the authors provide a concentration metric for alarm cue (cuts per liter – as per Chivers et al.2014) so that the results can be compared to other studies. This would help to compare the results found to other studies in the discussion (see lines 324-330 in discussion)
o Some of the methods are repeated or elaborated on in different sections – I suggest the authors review the manuscript for these repeats and maintain consistency for clarity and flow (example: lines 141-145 are similar to information provided in the previous paragraph).
o Throughout the methods, the authors should keep in mind that ‘N’ usually refers to population size while ‘n’ refers to sample size – I suggest the authors make appropriate changes throughout.
o Line 179-180: I would suggest the authors provide more clarity as to how they characterized a “bite” and make the association between “aggression” and “number of bites” more explicit for clarity.
o I am not sure if the method used to “increase power” is appropriate (lines 219-222) – is this a commonly used statistical method/experimental design? If so, I would suggest the authors include a reference or describe in more justification for the use of this method.
- Results:
o It is not clear whether lines 242-243 are referring to a statistical test or stating a trend in the data – I would suggest the authors clarify this statement.
- Discussion:
o References are needed throughout (for example the statement on lines 309-311 needs a reference) – I would suggest the authors review the manuscript entirely to make sure statements are referenced properly.

Experimental design

- While I think the basis of the study is interesting and important, I am not convinced by the results/statistical analyses used – particularly in relation to the repeated measure being collapsed for analysis. I would suggest the authors provide more support/justification for the method for further review.
- While the authors do a good job at describing the study system – I think the introduction could be strengthened by providing mention to an explicit knowledge gap they are addressing and providing some more context/examples from the literature relating to perceived predation risk and aggressive/social behaviour.
- As mentioned above (In the basic reporting) – some key issues have been identified relating to the clarity of methods described for replicability.

Validity of the findings

- The authors may have found some interesting results here (however, see issues above with statistical methods), but some of the claims in the discussion might be overreaching the limits of the experimental design and results presented
o For example, lines 255-257 suggest that “Amazon mollies respond to predation in similar ways as most sexually-reproducing small, shoaling fishes…” however, this study did not compare molly responses to sexually-reproducing fish responses.
- As mentioned above (results section) I think it is important for the authors to provide more discussion relating to the results found (for example line 246-247 results are not explicitly discussed)
- I found the discussion somewhat difficult to follow – I suggest the authors discuss the results one by one and explicitly relate the discussion to results/findings. For example, it is unclear, to me, how the paragraph from line 258-281 relates to the results found.
- The paragraph from line 282-296 suggests that the results from the repeated measure show a decline in sociability over time – this brings up a concern that the collapsing of the repeated measure for the more general model may not be a sound method for analyzing the data (to bolster power). If the authors are convinced that it is sound, I would suggest they provide more justification.

---

## Round 0.2 · accepted · Accept

Dear Jonathan et al,

Thank you for providing a clear and comprehensive response to the reviewer comments, thoughtful revisions, and carefully annotated code that allowed me to easily understand your analysis. It made my decision easy.

All the best,
Andy